# Assessing Overweight, Obesity, and Related Risk Factors in 8–9-Year-Old Children in Szczecin, Poland

**DOI:** 10.3390/jcm13237478

**Published:** 2024-12-09

**Authors:** Irmina Gapińska, Ewa Kostrzeba, Joanna Ratajczak, Anita Horodnicka-Józwa, Dominika Raducha, Tomasz Jackowski, Mieczysław Walczak, Elżbieta Petriczko

**Affiliations:** 1Department of Pediatrics, Endocrinology, Diabetology, Metabolic Disorders and Cardiology of Developmental Age, Pomeranian Medical University, 71-252 Szczecin, Poland; 2Institute of Physical Culture Sciences, University of Szczecin, 71-065 Szczecin, Poland

**Keywords:** childhood obesity, obesity-prevention programs, obesity-related risk factors

## Abstract

**Background:** Childhood obesity is a significant public health issue. This study aimed to evaluate the prevalence of overweight and obesity among 8- and 9-year-old children in Szczecin, Poland, and unlike other studies to assess differences in nutritional status within a single urban agglomeration of nearly 500,000 residents, it was influenced by place of residence and socioeconomic status. **Methods:** The study included 4705 children (2327 boys and 2378 girls) aged 8 and 9 years, attending 50 primary schools in Szczecin (45 public, 5 private) across four districts: North, Right Bank, City Center, and West. Anthropometric measurements were used to calculate BMI. Overweight was diagnosed when BMI was ≥85th percentile but lower than the 95th percentile for age and sex, while obesity was when the BMI was ≥95th percentile. Data on parental BMI, education, and place of residence were also collected. **Results:** The prevalence of overweight was 16.6%, and obesity was 6.2%. Overweight was more common in girls, while obesity was more frequent in boys. Children attending special education schools and living in the Right-Bank district had the highest rates of overweight and obesity. Parental obesity and low parental education, particularly the father’s, were the strongest risk factors for excess body weight. **Conclusions:** Differences in childhood nutritional status exist even within the same city, influenced by residence and socioeconomic factors. Parental obesity, low education, living in the city’s rural districts, and attending special education schools are key risk factors. Enhanced prevention programs tailored to these factors are crucial to combat childhood obesity effectively.

## 1. Introduction

Obesity is defined as a pathologic increase of adipose tissue mass, which leads to disruption of human bodily functions [1]. The increasing incidence of excessive body weight in children is associated with a greater risk of diseases such as type 2 diabetes, arterial hypertension, metabolic-associated fatty liver disease, sleep apnea, and dyslipidemia [2]. Obesity is also a risk factor for depression and anxiety [3]. Poland belongs to the group of countries with medium to high prevalence of overweight and obesity in children and adolescents in comparison to other European countries [4], which indicates that while the issue is not yet as severe as in some countries with very high rates, it is significant enough to warrant immediate attention in public health policy and clinical practice. Public health policies such as implementing school-based nutrition programs, promoting physical activity, and addressing socio-economic disparities can help reduce childhood obesity. Clinically, regular screening, family-based interventions, and interdisciplinary care involving dietitians and psychologists can provide effective prevention and management strategies. The prevalence of overweight and obesity in children aged 7–18 in Poland is 18.8–24.6% in boys and 14.3–17.4% in girls, while the prevalence of obesity accounts for 4.3–8.8% in boys and 2.7–4.2% in girls [4]. These data suggest the need for performing intervention programs aiming to increase awareness about the consequences of excessive body weight, such as “The Brave Eight”, the study created in cooperation with and from the initiative of the city of Szczecin, fully funded by public resources. This study describes the results of the first edition of this program (screening based on anthropometrical measurements in elementary schools, collecting data about parental nutritional status, education, and place of residence). The first edition of the program was conducted in Szczecin in 2016–2018 and the second stage, not described in this study, involved medical intervention in the outpatient clinic of the clinical Pomeranian Medical University hospital for children with excessive body weight. The second and third editions of “The Brave Eight” were continued until 2022; results of the change in the nutritional status of adolescents after intervention are going to be described in further manuscripts. The program’s goal was to improve the health of children aged 8 and 9 years living in the Municipality of Szczecin by performing a complex health and education intervention in children with proven risks of civilization diseases. The exceptionality of this project lies in the fact that it involved a very large group (4705 children); each participant received an individual report on potential deviations from the norms in the performed tests and educational advice. In addition, it was the first such study conducted in the Pomeranian region.

The 21st-century lifestyle, characterized by increased screen time, highly processed foods, and less physical activity, has significantly contributed to the rise in childhood obesity. Unlike the past when outdoor play and home-cooked meals were more common, children today often lead sedentary lives with greater exposure to fast food and sugary snacks. This shift towards convenience and technology has made it more challenging for families to maintain healthy routines, amplifying the prevalence of obesity in children. The data from the literature state that children who have obese parents are more likely to be obese themselves by 70%, while the risk of obesity in children who have parents with normal body weight is only 10% [5]. The other well-known risk factor for overweight and obesity incidence in children is the parents’ educational status. According to studies, the lower the education level of the parents, the greater the risk of excess body weight status in their children [6,7]. It can result from the fact that parents with higher education levels undertake healthier dietary choices and have more active lifestyles [8]. The aim of this research was to assess overweight, obesity, and related risk factors in 8–9-year-old children in Szczecin, Poland. This study is unique in its comprehensive approach to assessing childhood obesity risk factors by evaluating differences in nutritional status within a single urban agglomeration of nearly 500,000 residents, influenced by place of residence and socioeconomic status on a significantly large study population. Unlike many studies that address singular factors, this research highlights the compounded impact of parental obesity and education levels, providing a detailed quantification of risk increases. This specificity not only sheds light on regional trends but also offers actionable insights for tailoring public health programs.

## 2. Materials and Methods

Szczecin is a city located in north-western Poland. It comprises an area of 300.55 km^2^, and the total population accounts for 401,907 people (data from 31 December 2019) [9]. The city is divided into four districts: Północ (North), Prawobrzeże (Right-Bank), Śródmieście (City Center), and Zachód (West) as presented in diagram 1. According to data from 2016, the biggest district is Prawobrzeże (149 km^2^, 49.6% of the city area). People declared living there make up 22.3% of the city’s population. The most populated districts are Zachód (30.9%) and Śródmieście (31.3%) [10]. Szczecin is traditionally divided into two districts, Lewobrzeże (Left Bank) and Prawobrzeże (Right Bank), by the Oder River. The Left Bank is characterized by the concentration of services and residential areas, located mainly in the districts of Śródmieście and Zachód, whereas the Right Bank is a center of providing resources for the people living in the direct surroundings of the city and is, historically, an industrial district [11]. Data about population per 1 km^2^ of total surface according to districts were presented in Figure 1. Dividing the study by districts ensures representation of children from all parts of Szczecin, capturing nutritional disparities influenced by geographic, demographic, and socio-economic differences. This approach minimizes selection bias, enhances generalizability, and aligns with local institutional frameworks, facilitating data integration and policy development.

The study group consisted of the participants of the first screening stage of “The Brave Eight” program. Medical data for this analysis were gathered from 18 September 2016 to 31 December 2018. The target of the project was 8-year-old primary school students (11.494 children) born in 2008 (3939 children), 2009 (3792 children), and 2010 (3763), as well as their parents (caretakers). The inclusion criteria were age (8–9 years) and the consent of parents or legal guardians for participation in the study. The exclusion criterion was the absence of informed consent from a parent or legal guardian. 4705 out of 11,494 parents allowed their children to participate (40.93% of the population).

The mean age of the study population was 8.2 years, SD 0.64 (2327 boys, 2378 girls, *p* = 0.01). A total of 22.21% (1045) of children lived in the Right Bank part of the city and 77.79% (3660) in the Left Bank. Our study was screening-based, so we selected a group of school-aged children, which provided us with easy access and the opportunity to examine as many children as possible. Additionally, these were prepubertal children, allowing us to exclude the influence of sex hormones on increased insulin resistance. The selection of the age group was based on the belief that 8–9-year-olds constitute a group of children young enough that, through multidisciplinary intervention, we can influence their dietary habits and increase physical activity. At the same time, they are mature enough to provide an opportunity for effective collaboration. Children who repeated a grade or started education one year earlier were also invited to participate in the study so that they would not feel rejected by their peers. In order to minimize the risk of selection bias, the study targeted school-attending children to increase sample representativeness through easy access to participants, regardless of their nutritional status, background, or the type of school they attended. All schools in Szczecin decided to participate in the study. This study was performed in 45 public schools (3 special education schools—for children with physical or intellectual disabilities, 3 sport-profiled schools, and 1 music school) and 5 private schools, which were analyzed separately. The initial stage (screening) consisted of anthropometric measurements and a survey study. The questionnaire contained 26 questions concerning the child’s health, nutrition, frequency of the child’s physical activity, quality of sleep, fatigue, body weight at birth, the week of pregnancy at birth, and the education, height, and weight of the mother and father. Parents were offered anthropometric measurements in case they had any doubts about the current body weight and height. Body height was measured using a Harpenden type stadiometer with 1 mm accuracy, in standing (Frankfurt) upright position, without shoes. Every measurement was made three times, with the average result being calculated from all the measurements. The body weight was measured with 50 g accuracy in underwear or exercise clothes during physical education classes. BMI was calculated according to the formula $BMI=\frac{weight[kg]}{height[m^{2}]}$. Reference points used for body weight assessment were taken from the International Obesity Task Force (IOTF) for BMI [12]. The IOTF criteria are widely used in scientific research, facilitating data analysis. They are standardized at a global level, enabling comparisons of results across different populations and countries. This allows for better monitoring of obesity and overweight trends on an international scale. Overweight was diagnosed when BMI values were equal or higher than the 85th percentile but lower than the 95th percentile for age and sex, and obesity when the BMI was equal or higher than the 95th percentile. For the assessment of parents’ body weight, the following classification of nutritional status was applied: underweight—BMI < 18.5 kg/m^2^, normal value—BMI 18.5–24.9 kg/m^2^, overweight—BMI 25–29.9 kg/m^2^, obesity class I—BMI 30–34.9 kg/m^2^, obesity class II—BMI 35–39.9 kg/m^2^, and obesity class III—BMI ≥ 40 kg/m^2^ [13]. Statistical analyses were conducted using STATA 11 (license number 30110532736). Spearman’s rank correlation assessed relationships between numerical variables (body weight, height, and BMI), while the Chi-square test compared categorical data distributions (nutritional status classifications) across districts. Logistic regression evaluated the association between parental BMI (risk factor) and children’s nutritional status classifications (outcomes). Statistical significance was set at *p* < 0.05.

## 3. Results

Excess body weight was diagnosed in 22% (1072) of students in all schools involved in the program, in which 16% (780) were overweight and 6.2% (292) were obese. When divided by sex, overweight was diagnosed in 15.3% (352) of boys and 18% (428) of girls, whereas obesity was diagnosed in 7% (163) of boys and 5.4% (129) of girls. The results of the nutritional status classification have been graphically presented in Figure 2. There was a statistically significant difference between children’s nutritional status classification according to gender in all subtypes of nutritional status classification.

The largest proportion of students attended schools in Śródmieście (34.54%) and Zachód (31.97%), followed by Prawobrzeże (22.21%) and Północ (11.29%). The highest rates of overweight (19.33%) and obesity (8.8%) were observed in Prawobrzeże, while the other districts showed similar rates: Zachód (16.16% overweight, 5.45% obese), Śródmieście (15.51%, 5.11%), and Północ (15.63%, 6.59%). Spearman’s rank correlation analysis was used to examine relationships between numerical variables (body weight, height, BMI), and the Chi-square test compared the distribution of nutritional status classifications across districts. Detailed results are provided in Table 1.

The majority of the analyzed population (85.61%, 4028 students) attended non-profiled public schools, where 16.93% (701) were overweight. In contrast, 3.57% of the study group (168 students) attended sport-profiled schools, with lower rates of overweight 11.31%, 19) and obesity (2.98%, 5). Among the 159 students (3.38%) from special education schools, the prevalence of overweight was notably higher at 19.50% (31), and 7.55% (12) were classified as obese. Music school students represented the smallest subgroup, comprising only 3.0% (141 students) of the sample. Within this group, 11.35% (16) were overweight, while obesity was rare (0.71%, 1). Private school students accounted for 4.4% (209 students) of the total, with 12.44% (26) being overweight and 2.39% (5) classified as obese.

These findings highlight variations in BMI and nutritional status across different school types, with special education schools showing the highest prevalence of overweight and obesity.

Spearman’s rank correlation analysis was conducted to evaluate the strength and significance of the relationships between children’s BMI and the type of school attended. Additionally, a Chi-square test was applied to compare the distribution of nutritional status categories across different school types. Statistically significant results are highlighted in red. Results are presented in Table 2.

Children’s nutritional status was significantly associated with their parents’ BMI. Among mothers with normal body weight, 70.37% of their children also had normal weight, while being overweight and obesity were more prevalent among children of overweight or obese mothers. For instance, in mothers with class II obesity, only 44.16% of children had normal weight, while 28.57% were overweight and 23.38% were obese.

Similarly, children of overweight fathers had a higher likelihood of being overweight (16.8%) or obese (4.76%). Among fathers with class II obesity, 28.47% of their children were overweight and 18.06% were obese.

Spearman’s rank correlation analysis confirmed a positive relationship between parental and children’s nutritional status (R = 0.21 for mothers, R = 0.20 for fathers, *p* < 0.01 for both). The Chi-square test also showed significant differences in the distribution of children’s nutritional status across parental BMI categories (*p* < 0.01). Detailed data are presented in Table 3.

The analysis showed that parental overweight significantly increases the risk of children being overweight or obese. Maternal overweight increased the odds of children being overweight by 1.93 times and obese by 3.30 times. Similarly, paternal overweight increased the odds of children being overweight by 1.57 times and obese by 2.51 times.

Logistic regression analysis confirmed these associations, with odds ratios (ORs) greater than 1 indicating an elevated risk. All results were statistically significant (*p* < 0.01). Detailed data, including confidence intervals, are presented in Table 4.

The analysis revealed a strong association between parental BMI and the risk of children being overweight or obese, with maternal and paternal obesity showing particularly significant effects. Maternal obesity significantly increased the odds of both overweight and obesity in children, with a dose–response relationship observed. Mothers with class I obesity (BMI 30.0–34.9 kg/m^2^) had children with a 2.82 times higher risk of obesity compared to others. For class II obesity (BMI 35.0–39.9 kg/m^2^), this risk increased to 4.87 times, and for class III obesity (BMI > 40 kg/m^2^), it rose to 4.58 times.

Paternal obesity also had a substantial impact. Fathers with class II obesity increased the risk of obesity in their children by 3.99 times. Fathers with class III obesity had an even greater effect, raising the risk of obesity in children by 6.20 times.

The combined effect of both parents being overweight or obese was particularly striking. Overweight in both parents increased the risk of being overweight in children by 2.93 times. Obesity in both parents increased the risk of obesity in children by 9.61 times. These results were statistically significant (*p* < 0.01) and are detailed in Table 5. Logistic regression analysis confirmed that parental overweight and obesity are key risk factors for the child’s nutritional status. Detailed data are displayed in Table 5.

The analysis revealed significant associations between parental education levels and children’s nutritional status. Higher levels of parental education were linked to a greater proportion of children with normal body weight, while lower education levels were associated with higher rates of childhood overweight and obesity. Among mothers with higher education (2839), nearly 70% of their children had normal body weight, while 15.43% were overweight and 3.87% were obese. Fathers with higher education (2053) had children with a similar pattern, with 70.38% having normal body weight, 15.25% overweight, and only 2.58% obese. In contrast, mothers and fathers with primary education had higher proportions of children in overweight and obesity categories, with over 13% of their children being obese. Spearman Rank Correlation confirmed a weak but statistically significant negative association between parental education levels and unhealthy weight categories in children (R = −0.11 for mothers, R = −0.12 for fathers, *p* < 0.01). These results suggest that as parental education increases, the likelihood of children being overweight or obese decreases. Detailed data are gathered in Table 6.

Logistic regression analysis revealed significant associations between parental education levels and the risk of excess body weight (overweight or obesity) in children. Primary education of the father increased the risk of obesity in children by 2.46 times (OR = 2.46, *p* < 0.01), and in mothers, by 2.48 times (OR = 2.48, *p* < 0.01). Vocational education of the father was associated with nearly three times the risk of childhood obesity (OR = 2.92, *p* < 0.01), while in mothers, it doubled the risk (OR = 1.98, *p* < 0.01).

Secondary education increased the risk of obesity in children by 1.27 times for fathers (not statistically significant) and by more than twice for mothers (OR = 2.06, *p* < 0.01).

Higher education significantly reduced the risk of excess body weight in children. For fathers, it decreased the risk of obesity by 74% (OR = 0.26, *p* < 0.01) and overall excess body weight by 42% (OR = 0.58, *p* < 0.01). For mothers, it reduced the risk of obesity by 65% (OR = 0.35, *p* < 0.01) and overall excess body weight by 42% (OR = 0.58, *p* < 0.01).

These results underscore the protective effect of higher parental education against excess body weight in children, while lower education levels, especially primary or vocational education, pose a significant risk for childhood obesity. Detailed data are gathered in Table 7.

Zachód consistently showed the highest proportions of parents with higher education (69.89% of mothers, 53.49% of fathers), suggesting it has a more educated population. Prawobrzeże exhibited relatively lower levels of higher education and higher levels of vocational education, pointing to potential socioeconomic disparities. Fathers were more likely to have vocational education than mothers in all districts. Mothers had a slightly higher proportion of higher education compared to fathers across districts. A Chi^2^ Pearson Test revealed a statistically significant association between parental education level and city district (*p* < 0.01). Spearman’s Rank Correlation (R) indicated a weak but significant relationship between education levels and districts for both mothers (R = 0.04, *p* = 0.018) and fathers (R = 0.05, *p* = 0.00039). These trends highlight differences in parental education levels across city districts, which may reflect broader socioeconomic patterns. Detailed data are provided in Table 8.

## 4. Discussion

According to the WHO Childhood Obesity Surveillance Initiative, the number of children with obesity saw a dramatic increase between 1975 and 2016, rising from 5 million to 50 million in girls and from 6 million to 74 million in boys. The 2020 Global Nutrition Report highlights a significant rise in overweight incidence between 2000 and 2016, from 10.3% to 19.2% in boys and from 10.3% to 17.5% in girls. Similarly, the percentage of children with obesity nearly doubled, increasing from 3.3% to 7.8% in boys and from 2.5% to 5.6% in girls [14]. These data underscore that childhood obesity has become a severe and escalating global health crisis.

Data regarding the prevalence of overweight and obesity in the Pomeranian region from the past, as well as data regarding prevalence of excess body weight stats in the populations the most similar in terms of sociodemographic features (Germany, Norway), have been gathered in Table 9.

Our findings and the studies we cited suggest that overweight and obesity are becoming more common among Polish children. In our research, 16.6% of children were overweight, and 6.2% were obese. However, data cannot be easily compared due to differences in timing, research methods, socio-demographic features, and the age and size of the populations studied. We plan to analyze trends in overweight and obesity within our study group in future publications.

“The Brave Eight” program was the first large-scale health initiative in Western Pomerania, targeting a significant number of children and their caretakers. Out of 11,494 eligible children, 4705 (42.71%) participated. This study, focusing on the nutritional status of 8–9-year-olds in Szczecin, was unprecedented in scale and provided extensive data. It helped uncover the factors influencing childhood obesity in a region of approximately 500,000 residents. Unlike many studies that examine single factors, this research explored how parental obesity, education levels, and place of residence interact to shape obesity risk. These findings offer valuable insights for developing targeted obesity prevention programs. While these findings may not apply directly to other regions with different socio-economic or cultural contexts, the large sample size gives the results strong statistical reliability.

Childhood obesity is a significant socio-economic challenge, and identifying its causes is crucial for effective prevention strategies. A 2013 study by Januszek-Trzciąkowska et al. (*n* = 2571) found no association between childhood obesity and factors such as family income, parental education, or behaviors like TV watching, physical activity, or fast-food consumption, highlighting the critical role of diet in weight management [21]. In contrast, Weres et al. (*n* = 200, aged 3–6) demonstrated that maternal obesity is a significant risk factor for childhood obesity [22]. Notably, the increasing prevalence of obesity in children with slim parents suggests additional contributing factors, such as peer influence, dietary patterns, leisure activities, and social media exposure. While these factors were not assessed in the current study, they warrant exploration in future research. In Poland and worldwide, there is a noticeable decline in shared family meals, accompanied by frequent consumption of high-calorie snacks that parents often do not perceive as full meals. Children are also exposed to numerous food-related advertisements, spend significant time in front of screens, and engage in limited physical activity, particularly joint activities with their parents.

Krenc and Przybylska conducted a study examining the nutritional status of children at the beginning of their education in sport-profiled and non-profiled primary schools in Łódź. They found that overweight was more prevalent among children in non-profiled schools (13.02% vs. 7.32%), whereas obesity was more commonly diagnosed in students attending sport-profiled schools (14.64% vs. 2.17%). This phenomenon may be explained by parents enrolling their children in sport-focused schools to encourage a more active lifestyle and facilitate weight normalization [23].

Compared to these findings, overweight and obesity were more frequently diagnosed in children from public schools in the present study (17.01% and 6.56%, respectively). Children with intellectual disabilities are more likely to be underweight, whereas those with mild intellectual disabilities have a higher tendency for overweight and obesity, likely due to reduced spontaneous physical activity [24]. In this study, overweight was diagnosed in 17.96% and obesity in 8.74% of students from special education schools. However, these schools cater to children with diverse disabilities, making direct comparisons with the current results unreliable.

Obesity was least common among children attending music schools; among 141 students, only one case of obesity was identified (0.71%). In Poland, non-public schools are funded through tuition fees, suggesting that they are often chosen by parents with higher socioeconomic status.

The results of this study were based on data collected before the COVID-19 pandemic. Recent research highlights the pandemic’s negative impact on childhood obesity, primarily due to remote schooling, increased screen time, and reduced opportunities for physical activity [25]. For instance, a study from Indiana involving 27,093 participants (average age 9.8 years) reported a significant rise in obesity rates among children aged 5–11, with severe obesity increasing from 5.1% in 2019 to 6.3% in 2021, underscoring the pandemic’s lasting effects on children’s health [26]. The pandemic also exacerbated childhood obesity by increasing stress and limiting access to healthy foods, physical activity, and social routines—particularly for children in vulnerable communities affected by job losses, school closures, and restricted outdoor spaces [27].

Study limitations included reliance on self-reported parental BMI due to logistical constraints, as direct measurements were impractical given the large sample size and the school-based nature of the study coinciding with parents’ working hours. BMI’s limitations in assessing children’s body composition also posed challenges. However, overweight and obese children underwent additional evaluations, including bioimpedance body composition analysis, WHR/WHtR assessments, and a standardized physical fitness step test, as part of the second phase of “The Brave Eight” program.

The intervention stage of “The Brave Eight”, described by Raducha et al., involved 515 children who received specialist care for one year. At the end of the program, a statistically significant decrease in BMI Z-scores and WHR was observed. Interestingly, despite BMI improvement, many children exhibited persistent disruptions in carbohydrate and lipid metabolism [28].

These findings highlight the need for long-term programs with regular follow-ups targeting not only weight reduction but also education for children and their caregivers. The limited success of previous initiatives may be attributed to deeply rooted poor dietary habits among caregivers, which are often passed down from one generation to the next. From a socio-economic perspective, it is essential to expand and adapt these programs to reach a broader audience and achieve more lasting results.

## 5. Conclusions

1. The prevalence of overweight and obesity among 8–9-year-old primary school children in Szczecin was 16.58% and 6.21%, respectively. This rate aligns with the national average and reflects an upward trend observed in subsequent editions of “The Brave Eight” program.

2. The type of school, reflecting parental economic status, place of residence, and education level, significantly influenced the incidence of excess body weight in children.

3. The strongest predictor of childhood obesity was parental obesity and vocational education of the father. In contrast, higher education of both parents reduced the risk of childhood excess weight by 42%.

4. Given that paternal obesity and parental primary education were the most significant factors, effective interventions must target these groups. School-based and community programs should be prioritized to address childhood obesity through comprehensive prevention strategies that include educational outreach for families, particularly fathers with lower education levels. Based on the conclusions, preventive programs should prioritize the following goals:-Focus on family-centered education, integrating accessible materials and workshops tailored to low-education households.-Enhance school curriculums with regular health education sessions on nutrition and physical activity.-Encourage collaboration between schools and local community organizations to support families in creating healthier home environments.-Implement father-specific initiatives to engage and educate them about their critical role in shaping children’s health outcomes.-Track the effectiveness of ongoing interventions to provide follow-up of children’s nutritional status.

## Figures and Tables

**Figure 1 jcm-13-07478-f001:**
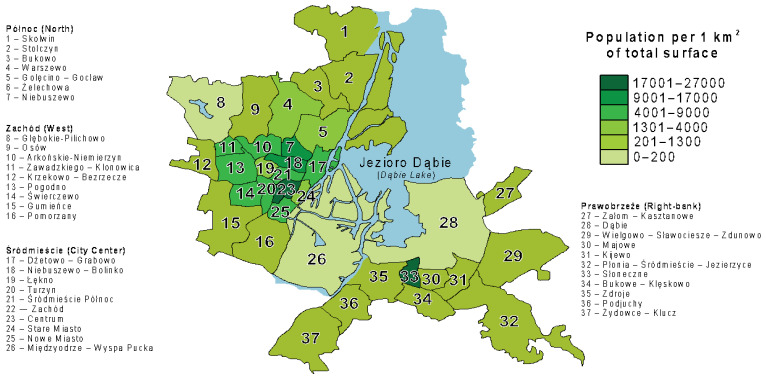
Szczecin according to districts and housing estates (data from 22 November 2017).

**Figure 2 jcm-13-07478-f002:**
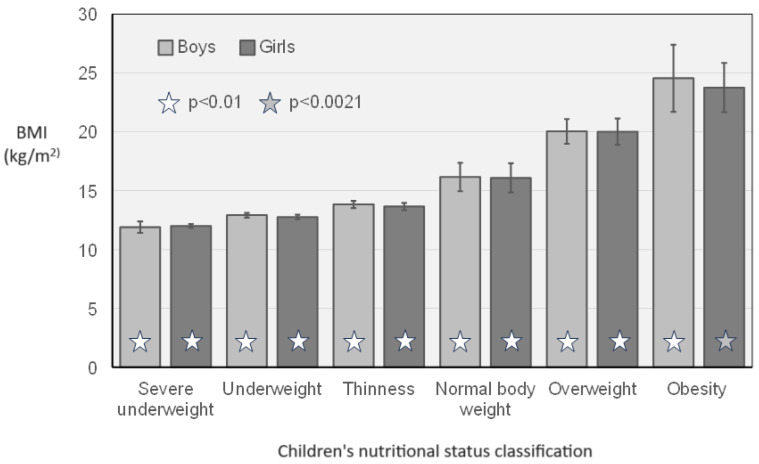
Children’s nutritional status classification based on BMI according to gender.

**Table 1 jcm-13-07478-t001:** Comparison of children’s auxologic data and nutritional status classification according to districts they inhabit. The data were compiled in one table below each other for technical reasons, as it was impossible to create a table that would compare all four districts in one line. Statistically significant correlations are marked in red.

Children’s Auxologic Data	Prawobrzeże (Right Bank) (*n* = 1045)				Północ (North) (*n* = 531)			
	Mean	SD	*p*	R	Mean	SD	*p*	R
Body weight (kg)	30.32	7.56	0.0032	0.04	29.81	6.98	0.8548	0.00
Body height (cm)	130.9	7.1	0.0207	0.03	131.6	6.7	0.3209	0.01
BMI (kg/m^2^)	17.42	3.11	0.0000	0.08	531	16.95	0.7172	0.00
**Children’s Auxologic Data**	**Zachód (West) (*n* = 1504)**				**Śródmieście (City Center) (*n* = 1625)**			
	**Mean**	**SD**	** *p* **	**R**	**Mean**	**SD**	** *p* **	**R**
Body weight (kg)	29.62	6.82	0.3881	0.01	29.49	6.89	0.0641	0.03
Body height (cm)	131.6	6.9	0.0816	0.02	131.3	6.8	0.7295	0.00
BMI (kg/m^2)^	16.87	2.74	0.0329	0.03	16.86	2.77	0.0147	0.04
**Children’s Nutritional Status Classification**	**Prawobrzeże (Right Bank)** **(*n* = 1045)**				**Północ (North)** **(*n* = 531)**			
Severe underweight	5 (0.48%)				1 (0.19%)			
Underweight	11 (1.05%)				12 (2.26%)			
Thinness	69 (6.6%)				53 (9.98%)			
Normal body weight	666 (63.73%)				347 (65.35%)			
Overweight	202 (19.33%)				83 (15.63%)			
Obesity	92 (8.8%)				35 (6.59%)			
Total	1045				531			
**Children’s Nutritional Status Classification**	**Zachód (West) (*n* = 1504)**				**Śródmieście (City Center) (*n* = 1625)**			
Severe underweight	2 (0.13%)				3 (0.18%)			
Underweight	29 (1.93%)				30 (1.85%)			
Thinness	149 (9.91%)				135 (8.35%)			
Normal body weight	999 (66.42%)				1122 (69.05%)			
Overweight	243 (16.16%)				252 (15.51%)			
Obesity	82 (5.45%)				83 (5.11%)			
Total	1504				1625			

χ^2^ = 0.00023. *p* = 0.0009.

**Table 2 jcm-13-07478-t002:** Children’s BMI and nutritional status by school type they attend.

Type of School		N		Children’s BMI					*p*		R	
				Average			SD					
**Public**		**4028**		**17.05**			2.90		* p * < 0.01		0.05	
Special education		159		17.41			3.24		0.0658		0.03	
Private		209		16.60			2.33		0.0421		0.03	
Music		141		16.11			2.12		* p * < 0.01		0.05	
Sport		168		16.54			2.17		0.0338		0.03	
**Children’s Nutritional Status Classification**	**Public**		**Special education**		**Private**	**Music**		**Sport**		**Total**		
Severe underweight	10 (0.24%)		0 (0.00%)		0 (0.00%)	0 (0.00%)		1 (0.60%)		11		χ^2^ = 0.00448 *p *= 0.00023
Underweight	70 (1.69%)		4 (2.52%)		4 (1.91%)	3 (2.13%)		1 (0.60%)		82		
Thinness	350 (8.45%)		15 (9.43%)		16 (7.66%)	19 (13.48%)		14 (8.33%)		406		
Normal body weight	2737 (66.10%)		97 (61.06%)		158 (75.60%)	102 (72.34%)		128 (76.19%)		3134		
Overweight	701 (16.93%)		31 (19.50%)		26 (12.44%)	16 (11.35%)		19 (11.31%)		780		
Obesity	273 (6.59%)		12 (7.55%)		5 (2.39%)	1 (0.71%)		5 (2.98%)		292		
Total	4141		159		209	141		168		4705		

**Table 3 jcm-13-07478-t003:** Parent’s nutritional status classification vs. children’s nutritional status classification and its frequencies.

Parents’ Nutritional Status Classification	Children’s Nutritional Status Classification						
	Severe Underweight	Underweight	Thinness	Normal Body Weight	Overweight	Obesity	Total
Mother							
Underweight	2 (1.31%)	9 (5.88%)	26 (16.99%)	103 (67.23%)	11 (7.19%)	2 (1.31%)	153
Normal body weight	6 (0.21%)	55 (1.88%)	276 (9.43%)	2059 (70.37%)	420 (14.35%)	110 (3.76%)	2926
Overweight	3 (0.33%)	10 (1.11%)	61 (6.76%)	539 (59.76%)	194 (21.51%)	95 (10.53%)	902
Class I obesity	0 (0.00%)	2 (0.67%)	15 (5.00%)	172 (57.33%)	68 (22.67%)	43 (14.33%)	300
Class II obesity	0 (0.00%)	2 (2.60%)	1 (1.30%)	34 (44.16%)	22 (28.57%)	18 (23.38%)	77
Class III obesity	0 (0.00%)	0 (0.00%)	1 (7.69%)	4 (30.77%)	5 (38.46%)	3 (23.08%)	13
Total	11	78	380	2911	720	271	4371
Chi^2^ Pearsona	263.36	df = 25	* p * < 0.01				
R rang Spearman	0.21	t = 13.963	* p * < 0.01				
Father							
Normal body weight	4 (0.33%)	38 (3.09%)	155 (12.61%)	859 (68.89%)	138 (11.23%)	35 (2.85%	1229
Overweight	4 (0.20%)	30 (1.54%)	152 (7.79%)	1345 (68.90%)	328 (16.8%)	93 (4.76%)	1952
Class I obesity	1 (0.15%)	7 (1.05%)	33 (4.97%)	409 (61.60%)	147 (22.14%)	67 (10.09%)	664
Class II obesity	0 (0.00%)	0 (0.00%)	6 (4.17%)	71 (49.31%)	41 (28.47%)	26 (18.06%)	144
Class III obesity	0 (0.00%)	1 (3.33%)	1 (3.33%)	14 (46.67%)	6 (20.00%)	8 (26.67%)	30
Total	9	76	347	2698	660	229	4019
Chi^2^ Pearsona	221.40	df = 20	* p * < 0.01				
R rang Spearman	0.20	t = 13.226	* p * < 0.01				

**Table 4 jcm-13-07478-t004:** The effect of parental overweight on children’s nutritional status classification.

Children’s Nutritional Status Classification	Mother					Father				
	Risk Factor	OR	[95%	CI]	*p*	Risk Factor	OR	[95%	CI]	*p*
Overweight	Overweight	1.93	1.63	2.29	* p * < 0.01	Overweight	1.57	1.31	1.87	* p * < 0.01
Obesity	Overweight	3.30	2.59	4.20	* p * < 0.01	Overweight	2.51	1.96	3.22	* p * < 0.01
Overweight and obesity	Overweight	2.61	2.24	3.04	* p * < 0.01	Overweight	1.99	1.70	2.32	* p * < 0.01

**Table 5 jcm-13-07478-t005:** The effect of parental BMI on children’s nutritional status classification.

Children’s Nutritional Status Classification	Risk Factor	Mother’s BMI [kg/m^2^]			OR	[95%	CI]	*p*
Overweight	Mother’s BMI	30.0–34.9	vs.	other	1.54	1.16	2.04	0.003
Obesity	Mother’s BMI	30.0–34.9	vs.	other	2.82	1.99	4.00	<0.01
Overweight and obesity	Mother’s BMI	30.0–34.9	vs.	other	2.13	1.67	2.72	<0.01
Overweight	Mother’s BMI	35.0–39.9	vs.	other	2.06	1.25	3.40	0.005
Obesity	Mother’s BMI	35.0–39.9	vs.	other	4.87	2.83	8.39	<0.01
Overweight and obesity	Mother’s BMI	35.0–39.9	vs.	other	3.80	2.42	5.98	<0.01
Overweight	Mother’s BMI	>40	vs.	other	3.18	1.04	9.76	0.043
Obesity	Mother’s BMI	>40	vs.	other	4.58	1.25	16.73	0.021
Overweight and obesity	Mother’s BMI	>40	vs.	other	5.49	1.79	16.83	0.003
Overweight	Father’s BMI	30.0–34.9	vs.	other	1.58	1.28	1.94	<0.01
Obesity	Father’s BMI	30.0–34.9	vs.	other	2.21	1.64	2.98	<0.01
Overweight and obesity	Father’s BMI	30.0–34.9	vs.	other	1.89	1.57	2.27	<0.01
Overweight	Father’s BMI	35.0–39.9	vs.	other	2.09	1.44	3.04	<0.01
Obesity	Father’s BMI	35.0–39.9	vs.	other	3.99	2.55	6.24	<0.01
Overweight and obesity	Father’s BMI	35.0–39.9	vs.	other	3.23	2.31	4.52	<0.01
Overweight	Father’s BMI	>40	vs.	other	1.27	0.52	3.13	0.596
Obesity	Father’s BMI	>40	vs.	other	6.20	2.73	14.08	<0.01
Overweight	Parents BMI	O or M > 24.9	vs.	O or M > 24.9	1.66	1.30	2.11	* p * < 0.01
Overweight	Parents BMI	O and M > 24.9	vs.	O and M > 24.9	2.93	2.26	3.82	* p * < 0.01
Obesity	Parents BMI	O or M > 24.9	vs.	O or M > 24.9	3.22	1.83	5.66	* p * < 0.01
Obesity	Parents BMI	O and M > 24.9	vs.	O and M > 24.9	9.61	5.47	16.88	* p * < 0.01

**Table 6 jcm-13-07478-t006:** The effect of parental education level on children’s nutritional status classification.

Parent’s Education	Children’s Nutritional Status Classification						
	Severe Underweight	Underweight	Thinness	Normal Body Weight	Overweight	Obesity	Total
Mother							
Primary	1 (0.81%)	2 (1.61%)	8 (6.45%)	81 (65.32%)	15 (12.10%)	17 (13.71%)	124
Vocational	2 (0.54%)	6 (1.63%)	30 (8.17%)	226 (61.58%)	63 (17.17%)	40 (10.90%)	367
Secondary	0 (0.00%)	15 (1.32%)	88 (7.72%)	694 (60.88%)	231 (20.26%)	112 (9.82%)	1140
Higher	8 (0.28%)	55 (1.94%)	257 (9.05%)	1971 (69.43%)	438 (15.43%)	110 (3.87%)	2839
Total	11	78	383	2972	747	279	4470
Chi^2^ Pearsona	106.39	df = 15	* p * < 0.01				
R rang Spearman	−0.11	t = −7.228	* p * < 0.01				
Father							
Primary	1 (0.74%)	1 (0.74%)	5 (3.68%)	93 (68.38%	18 (13.24%)	18 (13.24%)	136
Vocational	0 (0.00%)	13 (1.74%)	62 (8.32%)	441 (59.19%)	135 (18.12%)	94 (12.62%)	745
Secondary	4 (0.29%)	22 (1.60%)	107 (7.78%)	896 (65.12%)	250 (18.17%)	97 (7.05%)	1376
Higher	5 (0.24%)	43 (2.09%)	194 (9.45%)	1445 (70.38%)	313 (15.25%)	53 (2.58%)	2053
Total	10	79	368	2875	716	262	4310
Chi^2^ Pearsona	136.88	df = 15	*p* < 0.01				
R rang Spearman	−0.12	t = −8.005	*p* < 0.01				

**Table 7 jcm-13-07478-t007:** Statistical analysis of the parental education level as a risk factor for excess body weight in their children.

Children’s Nutritional Status Classification	Risk Factor by Parent	Level of Education			OR	[95%	CI]	*p*
Father								
Overweight	education	Primary	vs.	other	0.76	0.46	1.26	0.284
Obesity	education	Primary	vs.	other	2.46	1.47	4.10	* p * < 0.01
Overweight and obesity	education	Primary	vs.	other	1.24	0.84	1.82	0.286
Overweight	education	Vocational	vs.	other	1.14	0.92	1.40	0.224
Obesity	education	Vocational	vs.	other	2.92	2.24	3.81	* p * < 0.01
Overweight and obesity	education	Vocational	vs.	other	1.67	1.40	1.99	* p * < 0.01
Overweight	education	Secondary	vs.	other	1.18	0.99	1.39	0.060
Obesity	education	Secondary	vs.	other	1.27	0.98	1.65	0.068
Overweight and obesity	education	Secondary	vs.	other	1.23	1.06	1.43	* p * < 0.01
Overweight	education	Higher	vs.	other	0.83	0.70	0.97	0.022
Obesity	education	Higher	vs.	other	0.26	0.19	0.35	* p * < 0.01
Overweight and obesity	education	Higher	vs.	other	0.58	0.50	0.67	* p * < 0.01
Mother								
Overweight	education	Primary	vs.	other	0.68	0.39	1.17	0.165
Obesity	education	Primary	vs.	other	2.48	1.46	4.19	0.001
Overweight and obesity	education	Primary	vs.	other	1.17	0.78	1.76	0.444
Overweight	education	Vocational	vs.	other	1.04	0.78	1.38	0.807
Obesity	education	Vocational	vs.	other	1.98	1.39	2.82	* p * < 0.01
Overweight and obesity	education	Vocational	vs.	other	1.34	1.06	1.71	0.015
Overweight	education	Secondary	vs.	other	1.39	1.17	1.65	* p * < 0.01
Obesity	education	Secondary	vs.	other	2.06	1.61	2.65	* p * < 0.01
Overweight and obesity	education	Secondary	vs.	other	1.67	1.43	1.94	* p * < 0.01
Overweight	education	Higher	vs.	other	0.78	0.66	0.92	0.002
Obesity	education	Higher	vs.	other	0.35	0.27	0.45	* p * < 0.01
Overweight and obesity	education	Higher	vs.	other	0.58	0.50	0.66	* p * < 0.01

**Table 8 jcm-13-07478-t008:** Parent’s education level according to city districts.

Parent’s Education	District				
	Prawobrzeże	Północ	Zachód	Śródmieście	Total
Mother					
Primary	27 (2.72%)	20 (3.92%)	16 (1.11%)	61 (3.99%)	124
Vocational	99 (9.97%)	51 (10.00%)	69 (4.80%)	148 (9.88%)	367
Secondary	303 (30.51%)	131 (25.69%)	348 (24.20%)	358 (23.41%)	1140
Higher	564 (56.80%)	308 (60.39%)	1005 (69.89%)	962 (62.92%)	2839
Total	993	510	1438	1529	4470
Chi^2^ Pearsona	85.50	df = 9	* p * < 0.001		
R rang Spearman	0.04	t = 2.3636	* p * = 0.01814		
Father					
Primary	27 (2.83%)	22 (4.50%)	25 (1.78%)	62 (4.24%)	136
Vocational	204 (21.38%)	97 (19.84%)	178 (12.68%)	266 (18.18%)	745
Secondary	348 (36.48%)	154 (31.49%)	450 (32.05%)	424 (28.98%)	1376
Higher	375 (39.31%)	216 (44.17%)	751 (53.49%)	711 (48.60%)	2053
Total	954	489	1404	1463	4310
Chi^2^ Pearsona	81.70	df = 9	* p * < 0.001		
R rang Spearman	0.05	t = 3.5518	* p * = 0.00039		

**Table 9 jcm-13-07478-t009:** Comparison of data available in the literature regarding prevalence of overweight and obesity of the populations similar in terms of sociodemographic features.

Year of the Study	First Author	Age of the Study Population [Years]	Number of Study Population [n]	Location	Overweight Prevalence	Obesity Prevalence	Reference
2001	Małecka-Tendera et al. [15]	7–9	Polish and French population (2916 and 1582 respectively)	France, Poland	15.4% of Polish children, 18.1% of French children	3.6% of Polish children, 3.8% of French children	[15]
2007–2009	Kułaga et al. [4]	7–18	17,427	Different regions of Poland	18.8% of boys and 14.3% of girls	24.6% of boys, and 17.4% of girls	[4]
2007	Kurth et al. [16]	3–7	17,641	Germany	15%	6.3%	[16]
2012–2014	Kwilosz and Mazur [17]	7–11	1012	Bieszczady, Poland	10.4%	7.2%	[17]
2014	Milona et al. [18]	7–8	338	Szczecin, Poland	13.5% of boys and 13.6% of girls	8.6% of boys and 4.5% of girls	[18]
2018	Schienkiewitz et al. [19]	3–7	3561	Germany	15.4%	5.9%	[19]
2019	Meyer [20]	7–8	Not stated in the text	Norway	13%	4%	[20]

## Data Availability

The raw data supporting the conclusions of this article will be made available by the authors upon request.

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
