# Peer review of "Assessing Overweight, Obesity, and Related Risk Factors in 8–9-Year-Old Children in Szczecin, Poland"

_jcm, 2024, doi:10.3390/jcm13237478_

Round 1

Reviewer 1 Report

Comments and Suggestions for Authors

The article addresses a significant public health issue, childhood obesity, focusing on the nutritional status of 8-9-year-old children in Szczecin, Poland. The study is relevant due to the increasing prevalence of childhood obesity worldwide, which is linked to various health complications such as type 2 diabetes, hypertension, and metabolic disorders. This makes the article highly relevant for both medical and public health audiences.

The article is well-structured . It begins with an introduction outlining the importance of studying childhood obesity, followed by a detailed methodology section, results, discussion, and conclusions. The inclusion of well-defined subsections for each part of the research allows readers to follow the research process and findings clearly.

The study uses a large and representative sample of 4,705 children, which adds robustness to its findings. Anthropometric measurements and parental data were collected, and statistical analyses were performed to identify correlations between obesity and various factors such as parental body weight and education. The methods are adequately described, and the use of standardized BMI criteria based on the International Obesity Task Force (IOTF) ensures comparability with other studies.

The presentation of results is supported by numerous tables and figures, providing a clear and visual representation of the data.

The article references both recent and older studies, with citations dating from 2005 to 2020. While the inclusion of historical data helps establish trends in obesity prevalence, the article could benefit from including more recent sources, especially post-pandemic studies that explore the potential impact of COVID-19 on childhood obesity. The cited literature is relevant and supports the study's claims, but the inclusion of more recent meta-analyses or systematic reviews would strengthen the modernity of the research.

The discussion is thorough and relates the study's findings to existing research on childhood obesity, drawing comparisons with national and international data. The conclusions are well-supported by the results, emphasizing the role of parental obesity and educational level in childhood obesity. The authors also suggest practical implications, such as the need to target prevention programs at fathers with lower educational levels, which is a valuable insight for public health interventions.

The article is a valuable contribution to the understanding of childhood obesity in Poland. Its large sample size and detailed analysis provide meaningful insights into the factors contributing to obesity in children. However, it could be further improved by incorporating more recent literature on the subject.

Comments on the Quality of English Language

The English language in the article is understandable, and the main ideas are conveyed quite clearly. However, there are a few instances where the phrasing sounds somewhat awkward or not entirely natural for native speakers. The article contains minor grammatical errors, such as unnecessary commas, incorrect verb tenses, or subject-verb agreement issues. For example, in one sentence, the phrase "was diagnosed with obesity 5.5 times" is used— it would sound more natural to say "increased the risk of obesity by 5.5 times." The language sometimes sounds more formal or technical than necessary for a scientific article. This can make the reading more difficult, so it would be helpful to simplify some sentences and make them more concise. Some sentences are too long and complex, which may confuse the reader. For instance, it would be beneficial to split longer sentences into shorter ones to improve readability. The terminology used is mostly appropriate, but there are instances where word choice could be improved for greater accuracy or ease of understanding. For example, replacing some complex phrases with more familiar or natural alternatives for native speakers. Some statements could be made more precise or clearer. For instance, instead of saying "it is crucial to increase the number of prevention programs," it might be better to say "there is a need to increase the number of prevention programs." The article is well-written, but there are a few areas where the English language could be improved. To reach a high standard of writing, it is recommended to consult a linguist to correct stylistic, grammatical, and lexical errors.

Author Response

Dear Reviewer,

I am writing to express my sincere gratitude for the thorough review of our manuscript titled “The assessment of nutritional status of 8-9-year-old children residing in Szczecin (Poland) with specific focus on prevalence of overweight and obesity along with an evaluation of selected obesity-related risk factors” Your insightful comments and constructive suggestions have been invaluable in enhancing the quality of our work. In response to your feedback, we have made numerous revisions to address the concerns raised during the review process. All changes in the manuscript have been marked in red to facilitate tracking. Here is a summary of the key modifications made to the manuscript.

Comment 1: The article addresses a significant public health issue, childhood obesity, focusing on the nutritional status of 8-9-year-old children in Szczecin, Poland. The study is relevant due to the increasing prevalence of childhood obesity worldwide, which is linked to various health complications such as type 2 diabetes, hypertension, and metabolic disorders. This makes the article highly relevant for both medical and public health audiences.

The article is well-structured . It begins with an introduction outlining the importance of studying childhood obesity, followed by a detailed methodology section, results, discussion, and conclusions. The inclusion of well-defined subsections for each part of the research allows readers to follow the research process and findings clearly. The study uses a large and representative sample of 4,705 children, which adds robustness to its findings. Anthropometric measurements and parental data were collected, and statistical analyses were performed to identify correlations between obesity and various factors such as parental body weight and education. The methods are adequately described, and the use of standardized BMI criteria based on the International Obesity Task Force (IOTF) ensures comparability with other studies. The presentation of results is supported by numerous tables and figures, providing a clear and visual representation of the data. The article references both recent and older studies, with citations dating from 2005 to 2020. While the inclusion of historical data helps establish trends in obesity prevalence, the article could benefit from including more recent sources, especially post-pandemic studies that explore the potential impact of COVID-19 on childhood obesity. The cited literature is relevant and supports the study's claims, but the inclusion of more recent meta-analyses or systematic reviews would strengthen the modernity of the research. The discussion is thorough and relates the study's findings to existing research on childhood obesity, drawing comparisons with national and international data. The conclusions are well-supported by the results, emphasizing the role of parental obesity and educational level in childhood obesity. The authors also suggest practical implications, such as the need to target prevention programs at fathers with lower educational levels, which is a valuable insight for public health interventions.

The article is a valuable contribution to the understanding of childhood obesity in Poland. Its large sample size and detailed analysis provide meaningful insights into the factors contributing to obesity in children. However, it could be further improved by incorporating more recent literature on the subject.

Response: Thank you for this valuable comment. According to the suggestion in the discussion section we have expanded on three post-pandemic studies that explore the impact of COVID-19 on childhood obesity (page 13-14, lines 379-388).

Comment 2: The English language in the article is understandable, and the main ideas are conveyed quite clearly. However, there are a few instances where the phrasing sounds somewhat awkward or not entirely natural for native speakers. The article contains minor grammatical errors, such as unnecessary commas, incorrect verb tenses, or subject-verb agreement issues. For example, in one sentence, the phrase "was diagnosed with obesity 5.5 times" is used— it would sound more natural to say "increased the risk of obesity by 5.5 times." The language sometimes sounds more formal or technical than necessary for a scientific article. This can make the reading more difficult, so it would be helpful to simplify some sentences and make them more concise. Some sentences are too long and complex, which may confuse the reader. For instance, it would be beneficial to split longer sentences into shorter ones to improve readability. The terminology used is mostly appropriate, but there are instances where word choice could be improved for greater accuracy or ease of understanding. For example, replacing some complex phrases with more familiar or natural alternatives for native speakers. Some statements could be made more precise or clearer. For instance, instead of saying "it is crucial to increase the number of prevention programs," it might be better to say "there is a need to increase the number of prevention programs." The article is well-written, but there are a few areas where the English language could be improved. To reach a high standard of writing, it is recommended to consult a linguist to correct stylistic, grammatical, and lexical errors.

Response: Thank you for this noteworthy suggestion. The text has been revised for stylistic, grammatical and lexical errors. We have completely revised the text to include as much specific information as possible, presented in a way that is accessible to the reader.

Thank you once again for your valuable contribution to the enhancement of our manuscript. I believe that your comments have not only improved the clarity of the presentation but have also contributed significantly to the overall strength of the research. Your guidance has been

instrumental in refining the key aspects of the paper. Please feel free to reach out if further

clarification or information is required.

On behalf of all the co-authors

Yours sincerely

Ewa Kostrzeba

Reviewer 2 Report

Comments and Suggestions for Authors

Review Report (jcm-3281182)

Title: The assessment of nutritional status of 8-9-year-old children residing in Szczecin (Poland) with specific focus on prevalence of overweight and obesity along with an evaluation of selected obesity-related risk factors

Title and Abstract

- Please remove the “period” from your title after “obesity-related risk factors.”

- The abstract presents childhood obesity as a "major public health problem," which is well known. It follows a very conventional approach, focusing on the prevalence and risk factors already documented (parental education and weight), without pointing out new aspects or angles on these factors. Add a short sentence at the beginning that presents an innovative gap or that shows how this study stands out.

Introduction

- Although the authors mention childhood obesity as a risk factor for several diseases, the contextualization is limited, as it does not include more recent or global references on the impact of the pandemic or broader socioeconomic factors. The authors should also consider incorporating information on how recent lifestyle changes influence obesity in children.

- Factors such as parental obesity and educational level were listed as predictors, but these factors are already well known. I recommend that the authors organize the introduction to make clear the importance of investigating whether there are new combinations or contexts in which these factors interact differently. If these factors were analyzed in a unique way, this could be introduced here, to signal that the study adds something new to the literature.

- The authors suggest that Poland has a medium-high prevalence of childhood obesity. But what does this mean in terms of public health policy or clinical practice?

- The interest in investigating outcomes in 8- and 9-year-old children (specifically) as a target population is not justified. For example, would it be an early phase of transition in eating habits? Or another justification?

- The aim of the study is unconventional. Five lines of objectives, without breaks, does not seem appropriate. Please consider adapting the title to be more general.

Methods

- Please provide more details on the method of calculating and interpreting BMI, including references and justifications for the cutoff points.

- Please clarify the selection and definition of the exclusion and inclusion criteria to minimize bias.

- The methods are very objective. Please provide more information on the description of the recruitment procedures, emphasizing how selection bias was mitigated.

- What is the justification for using four districts as a spatial division? Is this division adopted by any local health or educational institution (Poland)?

- Explain the statistical analysis procedures in more detail.

Results

There are many tables in the manuscript. Tables 1 and 2 present auxological data of children by district (Prawobrzeże, PóÅ‚noc, Zachód and ÅšródmieÅ›cie) with few significant differences. Combine these data into a single table, focusing only on the variables that showed statistical significance. Table 3 can also be incorporated together with Tables 1 and 2. See an example of a Table that incorporates its Tables 1, 2 and 3 at the end of this report.

- Try to optimize the other Tables to make the manuscript more attractive.

- In Table 9, the effect of BMI should be described in a summarized way. The authors can focus on the most relevant BMI categories (obesity classes II and III) and omit less significant details for the weight of the children.

Discussion

- Review the first paragraph of the discussion. It should focus on the main results of the study in light of the objectives initially proposed. In addition, highlight the practical implications of these results and, if possible, indicate where exactly your study advances in light of the associations found.

- Authors should review the discussion to better contextualize their highlights in the context of similar international studies, highlighting cultural and, if possible, methodological differences.

- What theories about child eating behavior, emphasizing the impacts of parental influences, could be used to explain the parent versus child result?

- Some limitations of the study were omitted: i) the study does not consider variables such as specific eating habits, screen time and level of physical activity outside of school, which are critical factors for childhood obesity and could influence the results; ii) since it is a cross-sectional study, it is not possible to establish causal relationships between risk factors (such as parental obesity and parental education) and obesity in children; iii) data on parental weight and education were probably self-reported. Could this introduce response bias, especially in the case of information on parental BMI and family habits? iv) I personally like BMI very much, although I recognize its limitations. Please note that BMI is limited in assessing body composition, especially in children. Additional measures such as waist circumference or body fat percentages should be incorporated in future studies to provide a more accurate view of overweight; v) The study was conducted in a single city (Szczecin, Poland). Do the authors agree that the results may not be fully applicable to other regions, especially in different socioeconomic or cultural contexts?

Conclusions

- Avoid redundant phrases. If possible, the authors should reinforce the importance of school and community interventions to address childhood obesity. In addition, the authors should go further and objectively add practical recommendations for preventive programs based on family education.

-What future lines of research to monitor the effectiveness of ongoing interventions can be objectively indicated?

Author Response

Dear Reviewer,

I am writing to express my sincere gratitude for the thorough review of our manuscript titled “The assessment of nutritional status of 8-9-year-old children residing in Szczecin (Poland) with specific focus on prevalence of overweight and obesity along with an evaluation of selected obesity-related risk factors” Your insightful comments and constructive suggestions have been invaluable in enhancing the quality of our work. In response to your feedback, we have made numerous revisions to address the concerns raised during the review process. All changes in the manuscript have been marked in red to facilitate tracking. Here is a summary of the key modifications made to the manuscript.

Abstract

Comment 1: Please remove the “period” from your title after “obesity-related risk factors.”

The abstract presents childhood obesity as a "major public health problem," which is well known. It follows a very conventional approach, focusing on the prevalence and risk factors already documented (parental education and weight), without pointing out new aspects or angles on these factors. Add a short sentence at the beginning that presents an innovative gap or that shows how this study stands out.

Response: We appreciate this insightful comment. According to Comment 6 of this review we have changed the title for more general to “Assessing Overweight, Obesity and Related Risk Factors in 8-9-Year-Old Children in Szczecin, Poland” and changed the abstract – we included information about innovation of our study (assessing differences in nutritional status within a single urban agglomeration of nearly 500 000 residents, influenced by place of residence and socioeconomic status), elaborated on methodology, went over results and conclusions (page 1, lines 13-30).

Comment 2: Although the authors mention childhood obesity as a risk factor for several diseases, the contextualization is limited, as it does not include more recent or global references on the impact of the pandemic or broader socioeconomic factors. The authors should also consider incorporating information on how recent lifestyle changes influence obesity in children.

Response: Thank you for pointing this out. Because of limited word count in the abstract we have incorporated more data on the influence of the COVID-19 pandemic on the childhood obesity in the discussion section (page 13-14, lines 379-388), how recent lifestyle affects childhood obesity prevalence in the introduction (page 2, lines 67-27) and discussion (pages 12-13, lines 354-359).

Introduction

Comment 3: Factors such as parental obesity and educational level were listed as predictors, but these factors are already well known. I recommend that the authors organize the introduction to make clear the importance of investigating whether there are new combinations or contexts in which these factors interact differently. If these factors were analyzed in a unique way, this could be introduced here, to signal that the study adds something new to the literature.

Response: Thank you for pointing this out. This study is unique in its comprehensive approach to assessing childhood obesity risk factors by evaluating differences in nutritional status within a single urban agglomeration of nearly 500 000 residents, influenced by place of residence and socioeconomic status on a significantly large study population. Unlike many studies that address singular factors, this research highlights the compounded impact of parental obesity and education levels, providing a detailed quantification of risk increases. This specificity not only sheds light on regional trends but also offers actionable insights for tailoring public health programs. We have incorporated this information on page 2, lines 81-88.

Comment 4: The authors suggest that Poland has a medium-high prevalence of childhood obesity. But what does this mean in terms of public health policy or clinical practice?

Response: That is a valuable suggestion. Medium to high prevalence indicates that while the issue is not yet as severe as in some countries with very high rates, it is significant enough to warrant immediate attention in public health policy and clinical practice. Description of possible ways of reducing childhood obesity prevalence through public health policy and clinical practice was implemented on page 1-2, lines 40-46.

Comment 5: The interest in investigating outcomes in 8- and 9-year-old children (specifically) as a target population is not justified. For example, would it be an early phase of transition in eating habits? Or another justification?

Response: Thank you for pointing this out. Our study was screening-based, so we selected a group of school-aged children, which provided us with easy access and the opportunity to examine as many children as possible. Additionally, these were prepubertal children, allowing us to exclude the influence of sex hormones on increased insulin resistance. The selection of the age group was based on the belief that 8-9-year-olds constitute a group of children young enough that, through multidisciplinary intervention, we can influence their dietary habits and increase physical activity. At the same time, they are mature enough to provide an opportunity for effective collaboration. We elaborated justification of how we determined our target population within the Methodology section on page 3, lines 119-126.

Comment 6: The aim of the study is unconventional. Five lines of objectives, without breaks, does not seem appropriate. Please consider adapting the title to be more general.

Response: In accordance with the suggestion, we have changed the description of the aim of the study within manuscript to highlight its uniqueness and changed the title to “Assessing Overweight, Obesity, and Related Risk Factors in 8-9-Year-Old Children in Szczecin, Poland” (page 2, lines 79-88).

Methodology

Comment 7: Please provide more details on the method of calculating and interpreting BMI, including references and justifications for the cutoff points.

Response: Thank you for this comment. We have added the formula for calculating BMI and elaborated on justifications for the cutoff points (page 4, lines 144-155).

Comment 8: Please clarify the selection and definition of the exclusion and inclusion criteria to minimize bias.

Response: This is a valid point. We clarified the exclusion and inclusion criteria (page 3, lines 113-115).

Comment 9: The methods are very objective. Please provide more information on the description of the recruitment procedures, emphasizing how selection bias was mitigated.

Response: We appreciate this insightful comment. In order to minimize the risk of selection bias, the study targeted school-attending children to increase sample representativeness through easy access to participants, regardless of their nutritional status, background, or the type of school they attended (pages 3-4, lines 126-131).

Comment 10: What is the justification for using four districts as a spatial division? Is this division adopted by any local health or educational institution (Poland)?

Response: Organizing the study by districts ensures comprehensive representation of children across Szczecin, reflecting nutritional differences shaped by varying geographic, demographic, and socio-economic factors. This method reduces selection bias, improves the applicability of the findings, and aligns with local institutional structures, enabling effective data comparison and policy integration (page 2, lines 97-105).

Comment 11: Explain the statistical analysis procedures in more detail.

Response: The results section has been completely revised to present the findings in a more accessible and concrete manner, including a description of the statistical analysis.

Comment 12: There are many tables in the manuscript. Tables 1 and 2 present auxological data of children by district (Prawobrzeże, PóÅ‚noc, Zachód and ÅšródmieÅ›cie) with few significant differences. Combine these data into a single table, focusing only on the variables that showed statistical significance. Try to optimize the other Tables to make the manuscript more attractive. In Table 9, the effect of BMI should be described in a summarized way. The authors can focus on the most relevant BMI categories (obesity classes II and III) and omit less significant details for the weight of the children.

Response: Thank you for pointing this out. We have incorporated data from tables 1, 2 and 3, as well as put an effort to optimize other tables to make the manuscript more readable.

Discussion

Comment 13: Review the first paragraph of the discussion. It should focus on the main results of the study in light of the objectives initially proposed. In addition, highlight the practical implications of these results and, if possible, indicate where exactly your study advances in light of the associations found.

Response: That was a useful remark. Following the suggestion of the third reviewer, we have revised the entire Discussion section. We began by introducing data from other scientific studies on the prevalence of overweight and obesity in children. Then, in line with the comment, we emphasized how our findings stand out compared to those from other research studies (page 12, lines 334-341). In the Conclusions section, we described the practical implications of the results (page 14, lines 422-432).

Comment 14: Authors should review the discussion to better contextualize their highlights in the context of similar international studies highlighting cultural and, if possible, methodological differences.

Response: Results from our study have been compared to data obtained in Germany and Norway due to their proximity and sociodemographic resemblance to Poland (Table 9, page 12 lines 328-333).

Comment 15: What theories about child eating behavior, emphasizing the impacts of parental influences, could be used to explain the parent versus child result?

Response: Thank you for this comment. We have elaborated on those theories on pages 12-13, lines 351-359.

Comment 16: Some limitations of the study were omitted:

  1. the study does not consider variables such as specific eating habits, screen time and level of physical activity outside of school, which are critical factors for childhood obesity and could influence the results; Response: That’s a useful observation. In our study, we did not account for variables such as specific dietary habits, level of physical activity outside of school or screen time, but these are the factors with the great potential for exploration in the future research (pages 12-13, lines 351-359 as above). However, during the second phase of the first edition of the "The Brave Eight" program we assessed physical performance using a standardized step test and we plan to elaborate on that in future manuscripts (page 13, lines 389-395).
  2. since it is a cross-sectional study, it is not possible to establish causal relationships between risk factors (such as parental obesity and parental education) and obesity in children; Response: Spearman’s rank correlation analysis confirmed a positive relationship between parental and children’s nutritional status (R=0.21 for mothers, R=0.20 for fathers, p<0.01 for both) – Table 3. The analysis revealed a strong association between parental BMI and the risk of children being overweight or obese, with maternal and paternal obesity showing particularly significant effects – Table 5.
  • data on parental weight and education were probably self-reported. Could this introduce response bias, especially in the case of information on parental BMI and family habits? Response: In fact, the data on parental BMI were self-reported, as conducting direct measurements would have been extremely challenging due to the large sample size and the school-based nature of the study, which coincided with parents' working hours. Limitations of the study have been described on page 13, lines 389-395.
  1. I personally like BMI very much, although I recognize its limitations. Please note that BMI is limited in assessing body composition, especially in children. Additional measures such as waist circumference or body fat percentages should be incorporated in future studies to provide a more accurate view of overweight; Response: That’s a very good observation. For children with overweight and obesity, a second phase of the study was conducted, which included body composition analysis using the bioimpedance method as well as precise measurements of WHR and WHtR indices. Limitations of the study have been described on page 13, lines 389-395.
  2. v) The study was conducted in a single city (Szczecin, Poland). Do the authors agree that the results may not be fully applicable to other regions, especially in different socioeconomic or cultural contexts? Response: Yes, we agree with this statement. Nonetheless, even for a single city, the size of the study group was very large, providing statistical power to the obtained results (page 12, lines 342-344).

Conclusions

Comment 17: Avoid redundant phrases. If possible, the authors should reinforce the importance of school and community interventions to address childhood obesity. In addition, the authors should go further and objectively add practical recommendations for preventive programs based on family education.

Response: Thank you for this valuable suggestion. Based on that we have went over the Conclusions in order to highlight the importance of school and community interventions in childhood obesity prevention, as well as added practical recommendations for preventive programs tailored for the risk factors we have assessed.

Comment 18: What future lines of research to monitor the effectiveness of ongoing interventions can be objectively indicated?

Response: In order to monitor the effectiveness of ongoing interventions, we applied for an extension of the program's duration. Additionally, we designed a study to track the outcomes of children from the first edition of the project (which started six years ago), providing a follow-up analysis of their nutritional status during adolescence.

Thank you once again for your valuable contribution to the enhancement of our manuscript. I believe that your comments have not only improved the clarity of the presentation but have also contributed significantly to the overall strength of the research. Your guidance has been instrumental in refining the key aspects of the paper. Please feel free to reach out if further clarification or information is required.

On behalf of all the co-authors

Yours sincerely

Ewa Kostrzeba

Reviewer 3 Report

Comments and Suggestions for Authors

General comment

The paper aims to describe the prevalence of childhood overweight/obesity in Szczecin, Poland and to identify the main risk factors of childhood obesity in the studied region. The topic is interesting, since the prevalence of childhood overweightness and obesity is still increasing in several countries of the world, it is of high importance to get a more detailed information on influencing factors of obesity. The studied sample size is appropriate, huge.

As a general comment I like to emphasize that the presentation of the results is not appropriate (tables, text should be corrected, completed, statistical analyses are not introduced, why and when the analyses were used, the sample is not described appropriately, the subsamples are not compared whether it is possible to combine them etc.). The overall impression about the manuscript is that the preparation work was not precise enough, there are several inaccuracies both in the statistical methods and the presentation of the results.

The results of the analyses are not novel, it has been published from Europe and from the Eastern-Central European region that children nutritional status is related with parental nutritional status and parental education level.

I collected my comments and suggestions in the order of the sections of the manuscript to help the revision.

Abstract

A1: “… influence children body weight classification” – when the prevalence of obesity is studied, it is suggested to use the expression of nutritional status classification instead of weight classification, please correct the Abstract and the further parts of the manuscript by considering this suggestion.

A2: In the Methods part of the Abstract, no methods are introduced, only the sample. At least the nutritional status classification method should be mentioned in this section, please complete the Abstract.

A3: “Special education school students had the highest percentage of excess body weight.” – It is not mentioned in the Methods section in the Abstract how the excess of body weight was calculated (an ideal weight was estimated?), please complete the Methods section. Or the percentage of overweight/obese children is mentioned here, if so, correct the sentence, please.

A4: “The study found that primary education of the parent …” – which parent? Please complete the Abstract.

A5: “Given that the father's obesity and parents' primary education are key factors in childhood obesity …” – Father’s or both parents’ as it mentioned a few sentence before, parents’ primary education or only one of them as it mentioned a few sentence before? Please revise the Results section not to leave contradictory sentences.

A6: “to primarily target 30 dietary and lifestyle habits of the fathers with primary education.” – see A6 comment.

Keywords

K1: Why cardiometabolic risk is among the keywords? It is not analysed in the presented study, please find other keywords related to the presented study.

Introduction

I1: “The prevalence of overweight and obesity in children aged 7–18 is 18.8–24.6% in boys and 14.3–17.4% in girls, while the prevalence of obesity accounts for 4.3–8.8% in boys and 2.7–4.2% in girls.” – these data are for Poland or for Europe? Please complete the sentence.

I2: It is not clear why the Brave Eight program is mentioned in the Introduction section. if the Program has already shown results in improving children’s health status, nutritional status, please mention these results as well in the section. If not, describe the possibilities that the results of the prevention program appear in the studied sample, whether it is possible to see the decrease of childhood obesity in the studied sample due to the Brave Eight programmes between 2016-2022.

Materials and Methods

M1: “The study group consisted of the participants of “The Brave Eight” program.” Were they measured in 2024 or during the Brave Eight examinations? Make this clear, please! Base on the statement that “The mean age of the study population was 8.2 years.” – it is assumed that they were measured in different times and in different examinations during the different Brave Eight programmes. If so, please check and state that the subsamples did not differ by considering the prevalence of obesity and the other studied parameters.

M2: “Medical data for this analysis were gathered from 18.09.2016 to 31.12.2018.” – this is again not clear, based on Comment M2, please make it clear when and who were measured, how the subsamples can be described, why medical data were available on between 18.09.2016 and 31.12.2018.

M3: “Reference points used for body weight 109 assessment were taken from International Obesity Task Force (IOTF) for BMI.” – not for body weight, the mentioned IOTF references categorise the nutritional status by considering the BMI cut-off values, please correct the sentence. This comment goes to the complete manuscript (in tables and figures, too), where the Authors use “body weight classification”.

M4: “underweight BMI <18.5, normal value BMI 18.5–24.9, overweight BMI 25–29.9, obesity class I BMI 30–34.9, obesity class II BMI 35–39.9, obesity class III BMI ≥40 [13]” – please use the dimension of BMI (kg/m2) in the manuscript.

M5: “The test probability was considered significant with p<0.05.” – what kind of tests were used? Please describe them in this section, too.

Results

R1: Figures 1-2 are meaningless, please delete them, since BMI was used to divide children into nutritional status categories that are presented in these 2 figures.

R2: The type and name of the statistical test that was used to compare the districts are missing both in the text and in Tables 1 and 2, please complete the manuscript by giving details of the statistical analysis.

R3: It is not clear why “Prawobrzeże and PóÅ‚noc” and “Zachód and ÅšródmieÅ›cie” districts were compared in Tables 1-2, and why not all the comparisons were done, please complete the manuscript with the explanation or with the other comparisons.

R4: WHR was not mentioned before Table 1, please introduce it in the Materials an Methods section and mention it why WHR was used in the study.

R5: The statistical analyses are not described, it is not clear how could be stature, body weight, BMI, WHR with the same test studied (normal or not normal distribution?), or different tests were used for stature and the other variables?

R6: “It was observed, that the greatest number of students attended schools in ÅšródmieÅ›cie (City Centre) (1625 students, 34.54% of the studied population) and Zachód (West) (1504–138 31.97%), next Prawobrzeże (Right-Bank) (1045–22.21%) and PóÅ‚noc (North) (531–11.29%).” – wha this information is relevant, important in this manuscript?

R7: Only Table 3 shows that chi2 test was also used, please mention in the Material and Methods section.

R8: Table 4 should be revised – not to estimate the statistical parameter for boys and girls together, since BMI reveals sexual dimorphism in children, too. And do not estimate the average for BMI, it is not normally distributed, please use for example the media value. This means at the same time, not ANOVA or t test can be used to compare the BMI o subgroups, use non-parametric tests in the case of BMI.

R9: Which variable is presented in Table 6? BMI? Why not mentioned in the title? However, Table 6 is meaningless again, BMI is used to categorise parents into nutritional status categories, why to describe BMI in nutritional status subgroups? Delete Table 6.

R10: The title of the Table 7 is not correct, not only the correlation was studied if the frequencies of children/parents were also presented in the cells of the table, please complete the title.

R11: “Table 8. The effect of parental overweight on children’s body weight classification.” – the title is not correct, the effect of parental overweight not on nutritional status classification, but on nutritional status of children can be studied, please correct it. The same comment goes to Table 10.

R12: Table 8 suggests that a regression analysis was performed, but no details of the analyses is presented in the manuscript (which test?), please complete the manuscript. Same comment goes to the further tables.

Discussion

D1: Discussion section is too long. Make a Table to compare the different data from different studies, make the whole section shorter.

Conclusion

C1: “with visible upward trend observed in the next editions of “The Brave Eight” program.” – please revise this sentence to make it clear.

Author Response

Dear Reviewer,

I am writing to express my sincere gratitude for the thorough review of our manuscript titled “The assessment of nutritional status of 8-9-year-old children residing in Szczecin (Poland) with specific focus on prevalence of overweight and obesity along with an evaluation of selected obesity-related risk factors” Your insightful comments and constructive suggestions have been invaluable in enhancing the quality of our work. In response to your feedback, we have made numerous revisions to address the concerns raised during the review process. All changes in the manuscript have been marked in red to facilitate tracking. Here is a summary of the key modifications made to the manuscript.

General comment

The paper aims to describe the prevalence of childhood overweight/obesity in Szczecin, Poland and to identify the main risk factors of childhood obesity in the studied region. The topic is interesting, since the prevalence of childhood overweightness and obesity is still increasing in several countries of the world, it is of high importance to get a more detailed information on influencing factors of obesity. The studied sample size is appropriate, huge.

As a general comment I like to emphasize that the presentation of the results is not appropriate (tables, text should be corrected, completed, statistical analyses are not introduced, why and when the analyses were used, the sample is not described appropriately, the subsamples are not compared whether it is possible to combine them etc.). The overall impression about the manuscript is that the preparation work was not precise enough, there are several inaccuracies both in the statistical methods and the presentation of the results. The results of the analyses are not novel, it has been published from Europe and from the Eastern-Central European region that children nutritional status is related with parental nutritional status and parental education level. I collected my comments and suggestions in the order of the sections of the manuscript to help the revision.

Response: Thank you, we appreciate that. We put an effort to change the presentation of the results and fulfill data about statistical methods according to the instructions below. This study is unique in its comprehensive approach to assessing childhood obesity risk factors by evaluating differences in nutritional status within a single urban agglomeration of nearly 500 000 residents, influenced by place of residence and socioeconomic status on a significantly large study population. Unlike many studies that address singular factors, this research highlights the compounded impact of parental obesity and education levels, providing a detailed quantification of risk increases. This specificity not only sheds light on regional trends but also offers actionable insights for tailoring public health programs. We put this information on page 2, lines 81-88.

Abstract

A1: “… influence children body weight classification” – when the prevalence of obesity is studied, it is suggested to use the expression of nutritional status classification instead of weight classification, please correct the Abstract and the further parts of the manuscript by considering this suggestion.

Response: Thank you for this valuable comment. Every expression “body weight classifications” has been changed to “nutritional status classification” within all parts of the manuscript.

A2: In the Methods part of the Abstract, no methods are introduced, only the sample. At least the nutritional status classification method should be mentioned in this section, please complete the Abstract.

Response: That is a useful remark. We elaborated the methodology section including nutritional status classification.

A3: “Special education school students had the highest percentage of excess body weight.” – It is not mentioned in the Methods section in the Abstract how the excess of body weight was calculated (an ideal weight was estimated?), please complete the Methods section. Or the percentage of overweight/obese children is mentioned here, if so, correct the sentence, please.

Response: BMI was calculated according to the formula BMI=(weight [kg])/(height [m2]). Reference points used for body weight assessment were taken from International Obesity Task Force (IOTF) for BMI. We extended the Methodology on page 4, lines 144-160.

A4: “The study found that primary education of the parent …” – which parent? Please complete the Abstract. A5: “Given that the father's obesity and parents' primary education are key factors in childhood obesity …” – Father’s or both parents’ as it mentioned a few sentence before, parents’ primary education or only one of them as it mentioned a few sentence before? Please revise the Results section not to leave contradictory sentences. A6: “to primarily target 30 dietary and lifestyle habits of the fathers with primary education.” – see A6 comment.

Response: The abstract has been totally revised according to the suggestion (page 1, lines 13-30). Clarification of the influence of parental obesity has been made. We included information about innovation of our study, elaborated on methodology and went over results and conclusions for clarification.

Keywords

K1: Why cardiometabolic risk is among the keywords? It is not analysed in the presented study, please find other keywords related to the presented study.

Response: Thank you for correction, we have changed “cardiometabolic risk” for “obesity-prevention programs”.

Introduction

I1: “The prevalence of overweight and obesity in children aged 7–18 is 18.8–24.6% in boys and 14.3–17.4% in girls, while the prevalence of obesity accounts for 4.3–8.8% in boys and 2.7–4.2% in girls.” – these data are for Poland or for Europe? Please complete the sentence.

Response: These data concern the population of Polish children in 2016 – we have completed the sentence.

I2: It is not clear why the Brave Eight program is mentioned in the Introduction section. if the Program has already shown results in improving children’s health status, nutritional status, please mention these results as well in the section. If not, describe the possibilities that the results of the prevention program appear in the studied sample, whether it is possible to see the decrease of childhood obesity in the studied sample due to the Brave Eight programmes between 2016-2022.

Response: We are writing about the "The Brave Eight" program because the data for this study were collected as part of the first phase of this project. In various publications, we present different aspects of the collected data. In this particular paper, we focus on the prevalence of overweight and obesity and the differences related to place of residence and socioeconomic status, as this is an integral part of the program. In order to monitor the effectiveness of ongoing interventions, we applied for an extension of the program's duration. Additionally, we designed a study to track the outcomes of children from the first edition of the project (which started six years ago), providing a follow-up analysis of their nutritional status during adolescence. We have put that information within the manuscript on page 2, lines 51-60.

Materials and Methods

M1: “The study group consisted of the participants of “The Brave Eight” program.” Were they measured in 2024 or during the Brave Eight examinations? Make this clear, please! Base on the statement that “The mean age of the study population was 8.2 years.” – it is assumed that they were measured in different times and in different examinations during the different Brave Eight programmes. If so, please check and state that the subsamples did not differ by considering the prevalence of obesity and the other studied parameters. M2: “Medical data for this analysis were gathered from 18.09.2016 to 31.12.2018.” – this is again not clear, based on Comment M2, please make it clear when and who were measured, how the subsamples can be described, why medical data were available on between 18.09.2016 and 31.12.2018.

Response: Thank you for pointing this out. Medical data for this analysis were gathered from 18.09.2016 to 31.12.2018 as a part of the 1st screening stage of “The Brave Eight” programme. Children who repeated a grade or started education one year earlier were also invited to participate in the study so that they would not feel rejected by their peers, which is a reason for inviting children born in 2008, 2009 and 2010. We have tried to make it more transparent on page 3, lines 109-128.

M3: “Reference points used for body weight 109 assessment were taken from International Obesity Task Force (IOTF) for BMI.” – not for body weight, the mentioned IOTF references categorise the nutritional status by considering the BMI cut-off values, please correct the sentence. This comment goes to the complete manuscript (in tables and figures, too), where the Authors use “body weight classification”.

Response: Thank you for this noteworthy suggestion. “Body weight classification” was changed for “Nutritional status classification” within the whole manuscript.

M4: “underweight BMI <18.5, normal value BMI 18.5–24.9, overweight BMI 25–29.9, obesity class I BMI 30–34.9, obesity class II BMI 35–39.9, obesity class III BMI ≥40 [13]” – please use the dimension of BMI (kg/m2) in the manuscript.

Response: Thank you for this remark. We have applied dimension of BMI within the whole manuscript.

M5: “The test probability was considered significant with p<0.05.” – what kind of tests were used? Please describe them in this section, too.

Response: Thank you for this helpful input. We put an effort to totally change the presentation of the results and fulfill data about statistical methods.

Results

R1: Figures 1-2 are meaningless, please delete them, since BMI was used to divide children into nutritional status categories that are presented in these 2 figures.

Response: Figures 1 and 2 have been deleted.

R2: The type and name of the statistical test that was used to compare the districts are missing both in the text and in Tables 1 and 2, please complete the manuscript by giving details of the statistical analysis.

Response: Thank you for this insightful feedback. According to suggestions of the 2nd reviewer we have incorporated data within tables to reduce their number. We put an effort to fulfill data about statistical methods used.

R3: It is not clear why “Prawobrzeże and PóÅ‚noc” and “Zachód and ÅšródmieÅ›cie” districts were compared in Tables 1-2, and why not all the comparisons were done, please complete the manuscript with the explanation or with the other comparisons.

Response: The data was compiled in one table below each other for technical reasons, it was impossible to create a table that would compare all four districts in one line. We put that explanation on page 5, lines 196-197.

R4: WHR was not mentioned before Table 1, please introduce it in the Materials an Methods section and mention it why WHR was used in the study.

Response: Thank you for pointing this out. We have deleted WHR from this analysis, as the assessed correlations refer to the BMI value corresponding to the appropriate nutritional status, not WHR.

R5: The statistical analyses are not described, it is not clear how could be stature, body weight, BMI, WHR with the same test studied (normal or not normal distribution?), or different tests were used for stature and the other variables? R7: Only Table 3 shows that chi2 test was also used, please mention in the Material and Methods section. R12: Table 8 suggests that a regression analysis was performed, but no details of the analyses is presented in the manuscript (which test?), please complete the manuscript. Same comment goes to the further tables.

Response: Thank you for this helpful input. We did our best to organize the Results section and included a description of the statistical methods applied.

R6: “It was observed, that the greatest number of students attended schools in ÅšródmieÅ›cie (City Centre) (1625 students, 34.54% of the studied population) and Zachód (West) (1504–138 31.97%), next Prawobrzeże (Right-Bank) (1045–22.21%) and PóÅ‚noc (North) (531–11.29%).” – why this information is relevant, important in this manuscript?

Response: Thank you for this remark. In our opinion this is important information as it says about the size of the sample in individual districts and, consequently, describes the statistical significance of the results obtained, therefore we decided to keep them within the manuscript.

R8: Table 4 should be revised – not to estimate the statistical parameter for boys and girls together, since BMI reveals sexual dimorphism in children, too. And do not estimate the average for BMI, it is not normally distributed, please use for example the media value. This means at the same time, not ANOVA or t test can be used to compare the BMI o subgroups, use non-parametric tests in the case of BMI.

Response: Thank you for this valuable comment. In our opinion, prepubertal children do not exhibit such sexual dimorphism therefore, they were included together in the statistical analysis.

R9: Which variable is presented in Table 6? BMI? Why not mentioned in the title? However, Table 6 is meaningless again, BMI is used to categorise parents into nutritional status categories, why to describe BMI in nutritional status subgroups? Delete Table 6.

Response: Table 6 was deleted according to the suggestion.

R10: The title of the Table 7 is not correct, not only the correlation was studied if the frequencies of children/parents were also presented in the cells of the table, please complete the title. R11: “Table 8. The effect of parental overweight on children’s body weight classification.” – the title is not correct, the effect of parental overweight not on nutritional status classification, but on nutritional status of children can be studied, please correct it. The same comment goes to Table 10.

Response: The titles of the tables has been changed according to the suggestion.

Discussion

D1: Discussion section is too long. Make a Table to compare the different data from different studies, make the whole section shorter.

Response: Thank you for this  valuable comment. We have gathered data from different studies in the table 9, which indeed made them much more accessible for the reader and easier to compare.

Conclusion

C1: “with visible upward trend observed in the next editions of “The Brave Eight” program.” – please revise this sentence to make it clear.

Response: We have revised whole conclusion section according to suggestion of the 2nd Reviewer.

Thank you once again for your valuable contribution to the enhancement of our manuscript. I believe that your comments have not only improved the clarity of the presentation but have also contributed significantly to the overall strength of the research. Your guidance has been instrumental in refining the key aspects of the paper. Please feel free to reach out if further clarification or information is required.

On behalf of all the co-authors

Yours sincerely

Ewa Kostrzeba

Round 2

Reviewer 2 Report

Comments and Suggestions for Authors

None